# Effects of the COVID-19 Pandemic on the Behavioural Tendencies of Cats and Dogs in Japan

**DOI:** 10.3390/ani13132217

**Published:** 2023-07-06

**Authors:** Saho Takagi, Hikari Koyasu, Madoka Hattori, Takumi Nagasawa, Michiro Maejima, Miho Nagasawa, Takefumi Kikusui, Atsuko Saito

**Affiliations:** 1Department of Animal Science and Biotechnology, Azabu University, 1-17-71 Fuchinobe, Chuo-ku, Sagamihara 252-5201, Japan; h.koyasu.k@carazabu.com (H.K.); hattori.madoka.46p@st.kyoto-u.ac.jp (M.H.); nagasawa@carazabu.com (M.N.); kikusui@azabu-u.ac.jp (T.K.); 2Japan Society for the Promotion of Science, Kojimachi Business Center Building, 5-3-1 Kojimachi, Chiyoda-ku, Tokyo 102-0083, Japan; 3Wildlife Research Center, Kyoto University, 2-24 Tanaka-Sekiden-cho, Sakyo-ku, Kyoto 606-8203, Japan; 4Department of Animal Science, Graduate School of Agriculture, Tokyo University of Agriculture, Funako 1737, Atsugi 243-0034, Japan; cotaronagasawa@gmail.com; 5Graduate School of Human Sciences, Sophia University, 7-1 Kioi-cho, Chiyoda-ku, Tokyo 102-8554, Japan; 6Center for Human and Animal Symbiosis Science, Azabu University, 1-17-71 Fuchinobe, Chuo-ku, Sagamihara 252-5201, Japan; 7Department of Psychology, Faculty of Human Sciences, Sophia University, 7-1 Kioi-cho, Chiyoda-ku, Tokyo 102-8554, Japan

**Keywords:** pet ownership, Japan, COVID-19, mental health, cats, dogs, companion animal, behaviour, animal welfare

## Abstract

**Simple Summary:**

COVID-19 has had both physical and mental health effects stemming, in particular, from lockdowns. Furthermore, these physical and mental effects have impacted not only people’s lives but also the lives of their pets, which in recent years have often been seen as members of the family. This study aimed to investigate whether the COVID-19 pandemic affected pets and their owners in Japan. Therefore, we conducted an online questionnaire survey with cat and dog owners immediately after the state of emergency was announced in Japan. It comprised questions about owners’ physical activity and mental health, changes in their working conditions, frequency of interaction with pets, and pet behavioural tendencies before and after the COVID-19 lockdown, and then explored the relationships between them. Results showed that both cat and dog owners reported that the behavioural restrictions caused by the COVID-19 lockdown increased the amount of time they spent at home, thus increasing the frequency of contact interaction with their pets. Furthermore, we found that stress-related behaviours were higher in the pets of owners with increased contact interaction. The results of this study indicated that the behavioural restrictions of COVID-19 may have affected the lives of pets, their owners, and their relationships. The results obtained in this study provide important insights into the coexistence of humans and companion animals.

**Abstract:**

Physical and mental effects stemming from COVID-19 have impacted not only people’s lives but also the lives of their pets, which in recent years are often seen as members of the family. This study aimed to explore the impact of the COVID-19 pandemic in Japan on pets and their owners. Participants reported changes in physical activity and mental health, as well as working conditions and frequency of interaction with pets, before and after behavioural restrictions due to the COVID-19 pandemic. We also asked about their pets’ behaviours using the Feline Behavioural Assessment and Research Questionnaire (Fe-BARQ) and the Canine Behavioural Assessment and Research Questionnaire (C-BARQ). This study showed that most cat and dog owners spent more time at home due to the COVID-19 behavioural restrictions and that the frequency of contact interaction with their pets increased. However, this study showed higher stress-related behaviours (e.g., cats: excessive grooming; dogs: aggression towards owners) among pets whose owners increased contact interaction. Furthermore, owners’ low mental health was correlated with high stress-related behaviours (e.g., touch sensitivity) in pets. The results of this study indicate that the lockdown caused by the COVID-19 pandemic in Japan may have affected not only the lives of owners but also the interaction between owners and their pets, and consequently their pets’ behaviours. Therefore, there is also concern that changes in lifestyle patterns caused by pandemics could form a negative feedback loop between the health status of both owners and their pets.

## 1. Introduction

The COVID-19 outbreak, which started around 30 January 2020, affected people globally, causing a variety of problems [1]. Further, the effects of home confinement and social distancing on people have been studied worldwide [2] and have been reported to affect not only psychological aspects such as anxiety, depression, and increased stress [3], but also decreased physical activity [4]. Research reports on cats and dogs [5], the two most major companion animals, suggest that during the COVID-19 pandemic, owners may have gained psychological support from having an animal close by [6,7,8,9]. However, pet owners may have found factors such as the cost and care that come with pet ownership a burden [10,11,12,13,14]. In other words, no conclusions have yet been drawn about how the COVID-19 lockdown affects pet owners.

Similarly, COVID-19 lockdowns also affect the lives of pets. In dogs, regular walks with their owners were restricted [6,15], opportunities to see other dogs were reduced [15], and mental anxiety increased [16]. In addition, dogs’ demanding and separation anxiety-like behaviours [16,17], vocalization, and attention-seeking [6] increased. Moreover, the presence of people in the home also resulted in changes in the social environment, including decreased house-sitting and increased play/training opportunities [16]. In cats, COVID-19 lockdowns increased behaviours such as attention-seeking [6] and sociality [17]. In addition, cats either exhibited happy and relaxed appearances or behaved as if their owner’s presence was disturbing [7]. Thus, while it is clear that the COVID-19 pandemic has affected pets, there are no consistent conclusions about whether the effects are negative or positive.

Japan’s first lockdown was imposed from 7 April to 25 May 2020. The state of emergency declaration issued by the Japanese government in April [18] did not enforce curfews but was simply a “request” for people to refrain from going out, and there were no penalties if you did. On 27 February 2020, the Japanese government requested that all elementary, junior high, and high schools nationwide close temporarily, and as a result, 93% of public schools in Japan were closed as of 22 April after the state of emergency was declared [19]. In addition, as of 23 April, about 90% of universities had postponed the start of in-person classes [20]. However, there was no deterrent effect of these school closure requests on infection prevention [21].

As can be seen from the above studies, in Japan as in the rest of the world, the COVID-19 pandemic has brought about significant changes in people’s lives. The stay-at-home order implemented in Japan was designed to promote work-at-home and stay-at-home programmes, which were voluntary-type programmes free of penalties or other restrictions. Therefore, they were less enforceable than in other countries and are referred to as mild lockdowns [22,23]. Consequently, the impact of the COVID-19 pandemic on the Japanese population is likely to differ from that of other countries. 

With more than 15 million cats and dogs living in Japan [24], it is important to investigate the impact of COVID-19 on pet-owner relationships. For example, a study of Japanese university students showed that a high attachment to pets was a predictor of high positive affect [25]. In addition, compared to older adults who kept dogs, non-pet owners had lower psychological health scores [26]. Thus, it is clear that the presence of pets has a positive impact on Japanese pet owners’ mental health during COVID-19. However, it is unclear what changes occurred in pet owners’ lifestyles, relationships with their pets, and pet behaviour before and after COVID-19 in Japan and the state of emergency in Japan.

The purpose of this study was to investigate whether the lockdown associated with the spread of COVID-19 affected pets (cats and dogs) and their owners in Japan. We hypothesised that owners’ lifestyles and the frequency of communication between owners and their pets would change after lockdown. We assessed changes in owners’ lifestyles and health status (mental health and physical activity), frequency of owner-pet contact, and pet behaviour on the Feline Behavioural Assessment and Research Questionnaire (Fe-BARQ) [27] and Canine Behavioural Assessment and Research Questionnaire (C-BARQ) [28,29] after the first emergency declaration in Japan using an online survey and explored the relationship between these changes.

## 2. Materials and Methods

### 2.1. Participants

The questionnaires were collected online using social networking services (Twitter, Facebook, and Instagram) and by word of mouth starting July 2020. The criteria for taking the questionnaire included being a Japanese cat and/or dog owner, aged 20 or older. If participants had more than one cat or dog, they answered for one only. Responses to the questionnaire were obtained from 612 cat owners and 577 dog owners (See results and Table 1 for details).

### 2.2. Questionnaire

#### 2.2.1. Basic Information about the Participants and Changes in Their Working and Schooling Conditions Due to the COVID-19 Pandemic

This included sex, age, prefecture of residence, type of work (“full-time”, “part-time”, “student”, or “unemployed”), and condition of work or school before and after the COVID-19 pandemic (working at/from home, going to work/school, but the frequency decreased), educational background, household income, etc.

#### 2.2.2. Basic Information about the Cohabitants and Changes in Their Working and Schooling Conditions Due to the COVID-19 Pandemic

This included sex, age, relationship to the participant, type of work (“full-time”, “part-time”, “student”, or “unemployed”), and condition of work or school before and after the COVID-19 pandemic (working at/from home, frequency of going into work/school).

#### 2.2.3. Basic Information about the Cats/Dogs

This included the cat’s or dog’s sex, whether it was spayed or neutered, age, age at which they were first kept, breed and source, weight, and whether it was a multi-pet household. In addition, cat owners provided cat information on coat colour, eye colour, and hair length.

#### 2.2.4. Mental Health and Daily Physical Activity of the Participants

##### World Health Organisation Five Well-Being Index (WHO-5)

We used the WHO-5, developed by WHO, as a simple mental health indicator. The Japanese version of WHO-5 was developed by Awata et al. [30] after checking the equivalence of the original version and standardisation procedures [30]. The WHO-5 consists of five questions that ask participants about their mood in daily life. This has the advantage of being able to measure mental health over a short period of time.

The scoring was calculated by adding the five answer scores after converting them into numerical values as follows: always: 5 points; never: 0 points. The scores are from 0 to 25.

##### International Physical Activity Questionnaires-Short Form (IPAQ)

The short version of the IPAQ is a questionnaire designed to investigate physical activity in adults (15–69 years) [31]. The short version of the IPAQ scores walking, moderate physical activity, and strong physical activity using the number of days and times. In this study, we scored walking at 1 point, moderate physical activity at 2 points, and vigorous physical activity at 3 points, and then multiplied these scores by the number of days performing each activity, which was used as the amount of physical activity.

##### The Motor Fitness Scale (MFS)

The Japanese version of the MFS is a questionnaire that investigates the daily motor fitness of the elderly [32]. This scale has a unidimensional structure consisting of three subscales: mobility, strength, and balance. The answer to each item is “yes” (1 point) or “no” (0 point), and the total score ranges from 0 to 14 points. Although it was used to collect basic information for humans who own pets, it was not used for our analysis because this study included a few elderly people.

#### 2.2.5. Evaluation of the Human-Animal Interaction Frequency before and after the COVID-19 Pandemic

The participants were asked to indicate how often they approached, petted, called to, fed, or gave treats to their cats/dogs by choosing one of the following three options: increased, unchanged, or decreased. In addition, dog owners were asked about the frequency and duration of walks before and after the COVID-19 pandemic.

#### 2.2.6. Changes in the Temperament of the Cats/Dogs Due to the COVID-19 Pandemic

Changes in the temperament of cats and dogs were assessed using the Fe-BARQ [27] and C-BARQ [28,29], which assess behavioural characteristics. Participants were asked this both before and after the COVID-19 pandemic. These questions were assessed for the whole pre-COVID-19 pandemic and during the lockdown due to the COVID-19 pandemic, rather than for a specific day or the last two weeks.

The Fe-BARQ (for cats) and C-BARQ (for dogs), which are standardised questionnaires for assessing behavioural characteristics, were used. They were further subdivided into the following behavioural factors and traits. For Fe-BARQ, playfulness/activity, sociability (with people), directed calls/vocalization, purring, attention-seeking, sociability with cats, stranger-directed aggression, touch sensitivity/owner-directed aggression, resistance to restraint, familiar cat aggression, dog aggression, fear of unfamiliar dogs/cats, fear of novelty, separation-related behaviour, trainability, predatory behaviour, prey interest, location preferences for resting/sleeping, excessive/compulsive self-grooming, other compulsive behaviours, inappropriate elimination, elimination preferences, crepuscular activity, and other behaviours were assessed. For C-BARQ, trainability, owner-directed aggression, stranger-directed aggression, dog-directed aggression/fear, chasing, familiar dog aggression, stranger-directed fear, nonsocial fear, touch sensitivity, separation-related problems, excitability, attachment/attention-seeking, and energy level were assessed.

### 2.3. Analysis

Descriptive statistics presented the age, gender, number of cohabitants, age and gender of cohabitants, prefecture of residence, percentage of cohabitants whose lives have changed, number of children in cohabitants, whether children stay at home, the number of pets owned if the respondent owned more than one, and changes in the frequency of contact with cats and dogs. All analyses were conducted using R Studio (version 4.0.2).

We quantified whether the frequency of contact with cats and dogs increased due to the COVID-19 pandemic (change in contact score). In the question items for the frequency of the participant approaching, petting/touching, calling to, feeding, and giving treats to their pets, a point of +1 was given if the participant responded by saying that they increased, 0 if unchanged, and −1 if they decreased. The total score of these items was calculated as “change in contact score”. As a result, there were few people whose contact frequency decreased for both cats and dogs, and most of the respondents had a score of 0 or more. Therefore, respondents whose change in contact score was 0 or less were defined as the unchanged-decreased group and respondents with 1 or more as the increased group.

To examine whether the frequency of walking was changed by COVID-19 (for dogs only), the frequency of walking was given points: 5 for “twice a day”, 4 for “once a day”, 3 for “2 to 4 times a week”, 2 for “once a week”, 1 for “2 to 3 times a month”, and 0 for “hardly ever”. The duration (time taken) for a walk was also given points in the following manner: 4 for “more than an hour”, 3 for “30 min to an hour”, 2 for “15 min to 30 min”, and 1 for “less than 15 min”. 

To examine the effect of people’s presence at home on cats and dogs, participants who responded by saying “staying at home as before the state of emergency” were classified as the home/unchanged group, and those who responded by saying “went from commuting to work or school to being home due to the state of emergency” as the home/changed group, “commuted to work or school but the number of times decreased due to the state of emergency” as the office/changed group, and “went to work or school as before the state of emergency” as the office/unchanged group. Note that 19.76% (cats: 145 out of 612; dogs: 90 out of 577) of the survey respondents in this survey were single-person households, so only the duration that survey respondents stayed at home was analysed.

To examine the effect of changes in working conditions on mental health, the linear model (LM) was used, with working place (home/office), change by COVID-19 (changed/unchanged), and their interactions as the explanatory variables. The total score of the WHO-5 Well-Being Index was used for mental health. The lm function in the lmer package (version 1.1.10) was used for the model estimation. To test whether factor effects were significant, we ran F tests using an ANOVA function in the car package. The effect of changes in working conditions on physical activity was examined. Physical activity was calculated from the IPAQ. The LM was applied to the changing working place (home/office) change by COVID-19 (changed/unchanged) and their interactions as the explanatory variables and physical activity as the objective variables.

To examine whether contact with cats and dogs changed as a result of changes in working conditions, we conducted an LM with working place (home/office), change by COVID-19 (changed/unchanged), and their interactions as explanatory variables and change in contact score as the objective variable. 

To examine whether the working condition affects the frequency and duration of walks only in dogs, change by working place (home/office), change by COVID-19 (changed/unchanged), before or after (before/after), and their interactions were used as explanatory variables, frequency and duration of walks as objective variables, and individuals as random variables in an LMM (a Linear Mixed Model) using the lmer function in the lmer package (version 1.1.10). 

To examine the relationship between respondents’ mental health and increased contact with pets, a correlation analysis was conducted between the total mental health scores and changes in contact scores.

We analysed the relationship between physical activity and mental health. A correlation analysis between physical activity and mental health was conducted.

In the temperament items of Fe-BARQ and C-BARQ, LMM was conducted by using working place (home/office), change by COVID-19 (changed/unchanged), before and after (before/after), and their interactions as explanatory variables, and individuals as random variables.

To examine the effects on pets of the increased frequency of contact with cats and dogs due to the COVID-19 pandemic, we analysed the correlation between Fe-BARQ and C-BARQ in the increased and unchanged groups. For each temperament item, LMM was conducted using increased contact (unchanged-decreased/increased), before and after (before/after), and their interactions as explanatory variables, and individuals as random variables.

To examine whether survey respondents’ mental health changed their involvement with their pets, we analysed the relationship between Fe-BARQ and C-BARQ and respondents’ mental health. The total score of the WHO-5 Well-Being Index was used for mental health. For each temperament item, LMM was conducted by including mental health, before and after (before/after), and their interactions as explanatory variables and individuals as random variables.

All these tests were exploratory, and no α correction was made for the multiplicity of the tests.

## 3. Results

### 3.1. Descriptive Statistics

#### 3.1.1. Cat Results

The number of responses to the questionnaire was 678. After removing 66 duplicate responses, there were 612 valid responses. The gender of the survey respondents was overwhelmingly female (Table 1). The average age of the survey respondents was 45.35 years (*SD* = 10.36). Single-person households and two-person households were the majority (Table 1). The proportion of cohabitants who had changed their lifestyles was 46.35% (see Appendix A). Among 87 children, 53 stayed at home after the declaration of a state of emergency, and 30 saw no change (see Appendix A). There were 279 households with a single cat and 333 households with two or more cats (Table 1). There was a trend towards fewer respondents experiencing a decrease concerning the increase or decrease in each contact behaviour with a cat, with 345 people having a change in contact score greater than 0, 257 people having a score of 0, and 10 people having a score of less than 0 (Table 2). Cat information (age, sex, breed, etc.) is shown in the Appendix A.

#### 3.1.2. Dog Results

The number of responses to the questionnaire was 674. After eliminating 97 duplicate responses, the number of valid responses was 577. The gender of the survey respondents was overwhelmingly female (Table 1). The average age of the survey respondents was 48.41 years (*SD* = 9.82). Two-person households and three-person households were the majority (Table 1). The proportion of cohabitants who had changed their lifestyles was 43.97% (see Appendix A). Among 74 children, 56 stayed at home after the declaration of a state of emergency, and 10 saw no change (see Appendix A). There were 433 households with a single dog and 144 households with two or more dogs (Table 1). There was a trend towards fewer respondents experiencing a decrease concerning the increase or decrease in each contact behaviour with a dog, with 291 people having a change in contact score greater than 0, 278 people having a change in contact score of 0, and eight people having a change in contact score of less than 0 (Table 2). A total of 36 people increased the frequency of walks, 34 people decreased the number of walks, 491 people had no change, and 16 people were in the “other” category. A total of 40 people increased the duration of their walks, 46 decreased, and 491 had no change. The distribution of walking frequency and duration is shown in Table 3. Dog information (age, sex, breed, etc.) is shown in the Appendix A.

### 3.2. Relationship between Changes in Working Conditions and Mental Health

To examine the effect of working conditions on mental health, we analysed the relationship between two factors: working place (home/office) and change by COVID-19 (change/unchanged) and mental health.

#### 3.2.1. Cat Results

No differences in mental health were found in the four groups combining working place (home/office) and change by COVID-19 (change/unchanged) (Table 4): the main effect of working place (*F*(1, 608) = 0.037, *p* = 0.847), the main effect of change by COVID-19 (*F*(1, 608) = 1.468, *p* = 0.226), and interaction (*F*(1, 608) = 0.045, *p* = 0.831).

#### 3.2.2. Dog Results

No differences in mental health were found in the four groups combining working place (home/office) and change by COVID-19 (changed/unchanged) (Table 4): the main effect of working place (*F*(1, 573) = 0.022, *p* = 0. 881), the main effect of change by COVID-19 (*F*(1, 573) = 1.595, *p* = 0.207), and interaction (*F*(1, 573) = 1.407, *p* = 0.236).

### 3.3. Relationship between Changes in Working Conditions and Physical Activity

To examine the effect of changes in working conditions on physical activity, we analysed the relationship between two factors, working place (home/office) and change by COVID-19 (changed/unchanged) and the amount of daily physical activity.

#### 3.3.1. Cat Results

No differences in physical activity were found in four groups combining working place (home/office) and change by COVID-19 (changed/unchanged) (Table 5); the main effect of working place (*F*(1, 600) = 2.365, *p* = 0.125), the main effects of change by COVID-19 (*F*(1, 600) = 0.318, *p* = 0.573), and its interaction (*F*(1, 600) = 2.172, *p* = 0.141) were not significant.

#### 3.3.2. Dog Results

No differences in physical activity were found between the four groups combining working place (home/office) and change by COVID-19 (changed/unchanged) (Table 5). The main effects of working place (*F*(1, 568) = 0.041, *p* = 0.840), change by COVID-19 (*F*(1, 568) = 0.102, *p* = 0.749), and its interaction (*F*(1, 568) = 1.382, *p* = 0.240) were not significant.

### 3.4. Relationship between Changes in Working Conditions and Changes in Contact Scores for Pets

To examine whether the contact frequency with pets changed due to changes in working conditions, we analysed two factors: working place (home/office) and change by COVID-19 (changed/unchanged), and whether the contact frequency with companion animals increased.

#### 3.4.1. Cat Results

The main effects of working place (*F*(1, 608) = 8.350, *p* = 0.004) and change by COVID-19 (*F*(1, 608) = 99.165, *p* < 0.001) were significant. The home group had a higher change in contact scores than the office group, and the changed group had higher contact frequency change scores than the unchanged group; the interaction between working place and change by COVID-19 was not significant (*F*(1, 608) = 2.016, *p* = 0.156, Figure 1).

#### 3.4.2. Dog Results

The main effects of working place (*F*(1, 573) = 6.467, *p* = 0.011) and change by COVID-19 (*F*(1, 573) = 133.744, *p* < 0.001) were significant. There was also a significant interaction between working place and change by COVID-19 (*F*(1, 573) = 4.210, *p* = 0.041). Simple main effects were found to be higher in the home/changed group between home/changed and home/unchanged groups (*t*(573) = 9.359, *p* < 0.001), higher in the home/changed group between home/changed and office/unchanged groups (*t*(573) = 9.980, *p* < 0.001), higher in the office/changed group between office/changed and home/unchanged groups (*t*(573) = 6.457, *p* < 0.001), and higher in the office/changed group between office/changed and office/unchanged groups (*t*(573) = 7.097, *p* < 0.001), and higher in the home/changed group between home/changed and office/changed groups (*t*(573) = 3.214, *p* = 0.008, Figure 1).

### 3.5. Relationship between Changes in Working Conditions and Walking (The Dog)

To examine the effects of changes in working conditions on the frequency and duration of taking the dog for a walk, three factors were analysed in relation to the frequency and duration of walks: working place (home/office), change by COVID-19 (changed/unchanged), and before and after COVID-19 (before/after).

We examined whether or not there was a change in dog walking frequency and walking duration before and after COVID-19 in the four groups combining working place (home/office) and change by COVID-19 (changed/unchanged). The interaction between change by COVID-19 and before/after was significant for the frequency of dog walks (*χ*^2^(1) = 7.335, *p* = 0.007), but no significant differences were found for any of the comparisons in the subtests. The interaction between change by COVID-19 and before/after was also significant for walk duration (*χ*^2^(1) = 11.020, *p* < 0.001); however, the results of the subtests showed that none of the comparisons were significantly different (Table 6). 

### 3.6. Relationship between Respondents’ Mental Health and Contact Frequency with Pets

Pearson’s product-moment correlation test was conducted on the correlation between respondents’ mental health and contact frequency change scores.

#### 3.6.1. Cat Results

No correlation between mental health and contact frequency was found (*r* = −0.018, *p* = 0.651).

#### 3.6.2. Dog Results

No correlation between mental health and contact frequency was found (*r* = −0.025, *p* = 0.551).

### 3.7. Relationship between Physical Activity and Respondents’ Mental Health

Pearson’s product-moment correlation test was conducted on the correlation between respondents’ daily physical activity and mental health.

#### 3.7.1. Cat Results

There was a positive correlation between physical activity and mental health scores (*r* = 0.177, *p* < 0.001).

#### 3.7.2. Dog Results

There was a positive correlation between physical activity and mental health scores (*r* = 0.153, *p* < 0.001).

### 3.8. Effects of Changes in Working Conditions on Changes in Animal Temperament

We examined whether there was a change in the temperament of cats and dogs before and after the COVID-19 pandemic in relation to whether or not there were changes in where the owners worked primarily and the way they worked. Only those items for which there were significant differences were reported.

#### 3.8.1. Cat Results

LMM results showed a significant main effect of before/after on attention-seeking (*χ*^2^(1) = 5.059, *p* = 0.024). There were more owner-attractive behaviours in “after”. Trainability showed a significant interaction between working place and change by COVID-19 (*χ*^2^(1) = 8.341, *p* = 0.003). The results of the subtests showed significant differences between home/changed and office/changed (*t*(607) = 2.898, *p* = 0.020) and home/changed and home/unchanged (*t*(607) = 3.310, *p* = 0.005), with home/changed having a lower value in both. There was an interaction effect between working place and change by COVID-19 for location preferences for resting/sleeping (*χ*^2^(1) = 4.170, *p* = 0.042); however, the subtests showed no significant difference (Figure 2, Table 7).

#### 3.8.2. Dog Results

LMM results showed a significant main effect of change by COVID-19 on owner-directed aggression (*χ*^2^(1) = 4.904, *p* = 0.027). The changed group was higher than unchanged. In stranger-directed aggression, the interaction between working place and change by COVID-19 interaction was significant (*χ*^2^(1) = 7.954, *p* = 0.005). The results of the subtests showed that home/unchanged was higher between home/changed and home/unchanged (*df* = Inf, *z* = 2.750, *p* = 0.030), and office/unchanged was higher between home/unchanged and office/unchanged (*df* = Inf, *z* = 2.960, *p* = 0.016). In dog-directed aggression/fear, the main effect of change by COVID-19 (*χ*^2^(1) = 5.754, *p* = 0.016) and the interaction between working place and change by COVID-19 were significant (*χ*^2^(1) = 8.634, *p* = 0.003). The results of the subtests showed that home/unchanged was higher between home/changed and home/unchanged (*t*(588) = 3.785, *p* = 0.001), home/unchanged was higher between office/changed and home/unchanged (*t*(586) = 2.914, *p* = 0.019), and home/unchanged was higher between office/unchanged and home/unchanged (*t*(672) = 3.356, *p* = 0.004). In familiar dog aggression, the main effect of change by COVID-19 was found to be significant (*χ*^2^(1) = 4.716, *p* = 0.029). Changed was higher than unchanged. In stranger-directed fear, the main effect of working place (*χ*^2^(1) = 8.601, *p* = 0.003) and the interaction between working place and change by COVID-19 were significant (*χ*^2^(1) = 5.530, *p* = 0.018). The results of the subtests showed that home/unchanged was higher between home/changed and home/unchanged (*t*(598) = 2.790, *p* = 0.027), home/unchanged was higher between office/changed and home/unchanged (*t*(600) = 3.014, *p* = 0.014), and home/unchanged was higher between home/unchanged and office/unchanged (*t*(728) = 3.758, *p* = 0.001). In touch sensitivity, the main effect of change caused by COVID-19 was significant (*χ*^2^(1) = 4.745, *p* = 0.029). Changed was higher than unchanged. In energy, the main effect of working place (*χ*^2^(1) = 10.723, *p* = 0.001) and the interaction of working place on and change by COVID-19 were significant (*χ*^2^(1) = 7.954, *p* = 0.004). The results of the subtest showed that home/changed was higher between home/changed and home/unchanged (*t*(591) = 3.192, *p* = 0.008) and office/changed was higher between office/changed and home/unchanged (*t*(666) = 3.828, *p* < 0.001, Figure 3, Table 8).

### 3.9. Effects of the Presence or Absence of Changes in Contact Frequency on Changes in the Temperament of Pets

We examined whether there was a change in the temperament of cats and dogs before and after the COVID-19 pandemic in relation to increased owner contact. Only those items for which there were significant differences were reported.

#### 3.9.1. Cat Results

In playfulness/activity, the main effect of increased contact was significant (*χ*^2^(1) = 5.489, *p* = 0.019), with higher values for the unchanged-decreased groups; for attention-seeking, the main effect of increased contact (*χ*^2^(1) = 3.848, *p* = 0.049) and the main effect of before/after were significant (*χ*^2^(1) = 5.092, *p* = 0.024). The after and unchanged-decreased groups had a higher value. In fear of novelty, the main effect of increased contact was significant (*χ*^2^(1) = 17.977, *p* < 0.001), with higher values for the increased group. In excessive/compulsive self-grooming, the main effect of increased contact was significant (*χ*^2^(1) = 5.796, *p* = 0.016), with the increased group having higher values. In other compulsive behaviours, a main effect of increased contact was found, which was higher in the increased group (*χ*^2^(1) = 8.758, *p* = 0.003). In crepuscular activity, the main effect of increased contact was significant (*χ*^2^(1) = 10.297, *p* = 0.001), with higher values in the increased group (Figure 4, Table 9).

#### 3.9.2. Dog Results

In owner-directed aggression, the main effect of increased contact was significant (*χ*^2^(1) = 4.520, *p* = 0.034), with the increased group having a higher value. In familiar dog aggression, the main effect of increased contact was significant (*χ*^2^(1) = 7.844, *p* = 0.005), which was higher in the increased group. In nonsocial fear, the main effect of increased contact was significant (*χ*^2^(1) = 5.738, *p* = 0.016), which was higher in the increased group. In touch sensitivity, the main effect of increased contact was significant (*χ*^2^(1) = 8.232, *p* = 0.004), which was higher in the increased group. In attachment/attention-seeking, the main effect of increased contact was significant (*χ*^2^(1) = 13.927, *p* < 0.001), which was higher in the increased group. In energy, the main effect of increased contact was significant (*χ*^2^(1) = 5.204, *p* = 0.023), which was higher in the increased group (Figure 5, Table 10).

### 3.10. Impact of Owner’s Mental Health on Animal Temperament

We examined whether there was a change in the temperament of cats and dogs before and after the COVID-19 pandemic in relation to the owner’s mental health.

#### 3.10.1. Cat Results

In touch sensitivity/owner-directed aggression, the main effect of mental health was significant (*χ*^2^(1) = 4.614, *p* = 0.032). The lower the owner’s mental health, the higher the aggression towards the owner. In dog aggression, the main effect of mental health was significant (*χ*^2^(1) = 6.821, *p* = 0.009). The lower the owner’s mental health, the higher the aggression towards the dog. In separation-related behaviour, the main effect of mental health was significant (*χ*^2^(1) = 4.468, *p* = 0.035). The lower the owner’s mental health, the higher the value. There was no interaction between mental health and before/after for any of the items (Figure 6).

#### 3.10.2. Dog Results

In trainability, the main effect of mental health was significant (*χ*^2^(1) = 5.522, *p* = 0.019). The higher the mental health, the higher the trainability of the dog. In nonsocial fear, the main effect of mental health was significant (*χ*^2^(1) = 10.924, *p* < 0.001). The lower the mental health, the higher the nonsocial fear. In touch sensitivity, the main effect of mental health was significant (*χ*^2^(1) = 8.031, *p* = 0.005). The lower the mental health, the higher the touch sensitivity. In separation-related problems, the main effect of mental health was significant (*χ*^2^(1) = 9.403, *p* = 0.002). The lower the mental health, the higher the separation-related problems. In excitability, the main effect of mental health was significant (*χ*^2^(1) = 8.490, *p* = 0.003). The lower the mental health, the higher the excitability. There was no interaction between mental health and before/after any of the items (Figure 7).

## 4. Discussion

This study explored the effects of changes in people’s lifestyles on their physical activity, mental health, interaction with their cats and dogs, and their pets’ behaviour during the early stages of the COVID-19 pandemic in Japan.

### 4.1. Owners’ Mental Health and Interaction with Pets

Similar to findings in Western Asia, North Africa, and Europe [2], Japanese pet owners in this study experienced changes in their lifestyle before and after the lockdown. In addition, most of the pet owners who had experienced changes in their lifestyle reported an increase in the frequency of their involvement with their pets. This is consistent with previous reports [33,34,35]. A number of studies have reported the positive impact of animal presence on owners’ health states (e.g., well-being, quality of life, and loneliness) during COVID-19 lockdowns [6,7,8,9]. However, in the present study, there was no direct correlation between interaction with pets and owners’ mental health status. One reason for this is that the present study was a cross-sectional survey of owners’ mental health, and thus we were unable to observe changes before and after the lockdown due to COVID-19. In other words, the causal relationship between time spent with pets and owners’ health status is unclear. Another reason for this is that the COVID-19 restrictions were more relaxed compared to other countries. For example, in Australia, strict restrictions were imposed, such as limiting travel distance to 5 km or less [36]. In Japan, even the capital city of Tokyo only had a “mild lockdown” with no penalties for leaving the house; in other words, compared to other countries, the changes in interaction between pet owners and their pets caused by the COVID-19 behaviour restrictions in Japan were small and thus not necessarily related to the mental health of pet owners. However, some studies have reported negative effects of pet presence during the COVID-19 stay-at-home period on the owner’s health [10,11,12,13,14]. Further, animal ownership in people with severe mental illness was associated with deterioration in mental health [37], and the causal relationship between pet ownership and declining mental health needs to be carefully examined. Furthermore, Wells et al. reported that during the COVID-19 lockdown period, owners with higher attachments to their animals reported higher levels of depression and loneliness and fewer positive experiences [38]. This suggests a concern that pets can be both a protective factor for their owners’ health status and an object of excessive dependence for their owners.

### 4.2. Behaviours of Pets and Interaction with Their Owners

Although Fe-BARQ and C-BARQ are highly reliable scales to assess the behaviour of cats and dogs, respectively, they have not been utilised to assess pet behaviour under COVID-19 lockdowns. This is the first paper to utilise Fe-BARQ and C-BARQ to investigate the effects of COVID-19-induced changes in social life on the behaviour of Japanese pets. As a result, this study revealed that cats’ attention-seeking behaviours towards their owners increased before and after COVID-19. In addition, cats of owners whose frequency of contact interaction did not change before and after COVID-19 exhibited higher attention-seeking behaviours than cats of owners whose frequency of contact increased. Previous studies have also reported an increase in cats’ contact seeking [39], sociality [17], and attentional interest in humans [6] before and after COVID-19, which is consistent with the results of the present study. That being said, positive behavioural changes in pets have also been observed under the COVID-19 lockdown, such as relaxation and happiness in cats and dogs [7]. Cats and dogs have the ability to discriminate between different lengths of owner absence and pay attention to and show interest in interaction with their owners [40,41]. Furthermore, cats whose owners’ frequency of contact interaction did not change before and after COVID-19 were more playful and active during the day and less active at dusk and twilight, compared to cats whose owners increased contact frequency. Although no effect of changes before and after COVID-19 was found, the cats may have increased playful behaviour during the day in order to gain attention from their owners. These results indicate that cats may have increased their frequency of requesting interaction from their owners before and after COVID-19. However, the increase in owners’ time spent at home was also related to the trainability of their cats, with these cats showing lower trainability. In other words, although cats behaved in a manner to attract the attention of their owners, they did not show behavioural responses that were actually in line with their owners’ demands. This might be due to their highly independent nature, especially in contrast to dogs. Turner stated that for cats and humans to have a good relationship, humans need to accept cats’ highly independent temperament [42]. Indeed, previous studies have reported that communication initiated by the cat rather than the human is longer lasting [43]. This study suggests that although cats sense the presence of their owners and changes in communication and seek interaction with them, they seek interactions that they initiate themselves rather than from their owners.

This study found that dogs were more active when owners’ time spent at home increased due to COVID-19 lockdowns compared to dogs whose owners did not. Furthermore, cats with owners who increased contact with their cats showed less human attention-seeking behaviour than cats with owners who did not. Whereas dogs with owners who increased contact interaction showed more activity and attention-seeking behaviour towards their owners than dogs with owners who did not increase contact interaction. Dogs are highly social animals that pay more attention to their owners than cats do. Dogs are able to closely observe their owners’ behaviour and behave in sync with them [44]. Such interspecific synchrony plays an important function in the formation of social relationships between dogs and their owners; under COVID-19 lockdowns, dogs may have sensed signals of contact interaction desires and needs from their owners and selected behaviours to respond to them. However, no changes in the dog’s activity or attention-seeking behaviours were observed before or after the lockdown. Therefore, discussion of the effects of COVID-19 on pet behaviour, the owner’s lifestyle, and the pet-owner interaction relationship should proceed with caution.

Several studies have reported an increase in negative pet behaviours such as demanding and separation-related behaviour [16,17], vocalization, and attention-seeking [6] under COVID-19 lockdowns. In the present study, cats with owners who increased contact interaction were correlated with increased stress-related behaviours (fear of novelty, compulsive behaviours, and excessive/compulsive self-grooming) and decreased play and activity. As mentioned previously, cats tend to act more in pursuit of their own wants and needs than those of their owners. Therefore, increased demands for contact and interaction by owners may have been a stressor for cats. Furthermore, the increase in contact interaction may have restricted cats’ free behaviour during the day. Decreased cat play and hunting behaviour can be frustrating factors for cats [45]. Such changes in cat-owner communication and the house environment may act as stressors for cats [46]. Lockdown by COVID-19 also caused many changes in human consumption behaviour. In Japan, the frequency of spending on eating out and clothing decreased, while food hoarding behaviour and purchases of furniture and household goods increased [47]. In other words, the cats may have felt stressed by the increase in novel items in the home. In addition, the frequency of telework and online learning has also increased due to COVID-19 lockdowns [47], which may have increased the frequency with which cats are exposed to unfamiliar sound stimuli. These changes in the cat housing environment may have caused the cats to experience stress.

In dogs, aggression and fear towards strangers and other dogs were lower in dogs whose owners had increased their time spent at home than in dogs whose owners had not changed their time spent at home. This may be because their lifestyle changed, including the route and time of walks, the frequency of communication with other people and dogs, and a decrease in the frequency of visitors to the home. Further, for dogs with owners who increased their time spent at home, higher levels of aggression and touch sensitivity towards their owners and other dogs living with them were seen. In support of the present study, a study in Italy reported an increase in dog-biting incidents during the COVID-19 lockdown [48]. One explanation for this is the lack of social communication in dogs. For example, it has been reported that COVID-19 lockdowns reduced the opportunity to meet other dogs through walks [15] and that this increased mental anxiety [16]. For dogs, which have high sociability, limited communication with other dogs may be a strong stressor. Furthermore, dogs of owners with increased contact frequency showed higher aggression towards their owners, nonsocial fearfulness, and touch sensitivity. It is possible that dogs do not perceive contact interaction with their owners as a positive stimulus when social communication is lacking. The desire for social communication between dogs may not necessarily be substituted by that between humans and dogs. For example, owners may anthropomorphise their dogs and perform contact behaviours (e.g., hugging) that are stressful for some dogs [49]. Furthermore, the high stress state of dogs during the COVID-19 lockdown period may lead to dog bite accidents [48]. Understanding the differences in human and canine perceptions of communication may be important in managing canine stress. Owners should not increase the frequency of contact interaction to reduce their own stress in times of emergency but should understand the behavioural characteristics of cats and dogs and satisfy their frustrations appropriately. However, the stress-related behaviours of cats and dogs described above did not change statistically significantly from before to after the COVID-19 lockdowns. Therefore, the relationship between the effects of the COVID-19 pandemic on the lifestyle of pet owners and changes in pet behaviour remains unclear, and further research is needed.

### 4.3. Owners’ Mental Health and Behaviours of Their Pets

This study found that separation anxiety and touch sensitivity were higher in cats and dogs when their owners’ mental health was low. It is known that the health status of the owner and the pet is synchronised. For example, dog hair cortisol levels are positively correlated with owners’ hair cortisol levels [50,51], indicating that the long-term health status of dogs and their owners is synchronous. Moreover, even under the COVID-19 lockdown, owners’ well-being was associated with their dogs’ well-being [52], and pets’ separation anxiety is positively correlated with owners’ separation anxiety [53], which is consistent with the results of the present study. One hypothesis that explains this result is that the mental health status of the owner can be stressful for the pet. For example, it is known that stress can be passed on to dogs when they observe their owners in stressful situations [54]. In other words, it is conceivable that pets may have also been stressed by the owners’ stress caused by COVID-19. However, another hypothesis to explain this result is that cat and dog problem behaviours (e.g., separation anxiety [55,56]) negatively affect the mental health status of their owners. The problematic behaviour of pets is a major influencing factor in owners’ satisfaction and happiness with their pets [57,58,59] and is one of the reasons why owners abandon their pets [60,61]. Furthermore, in times of emergency, pets’ problematic behaviours can have a significant impact on the mental health of pet owners. Indeed, previous studies have shown that pets’ separation anxiety increased before and after the spread of the COVID-19 pandemic [16,17]. Owners are also concerned that their pets’ separation anxiety symptoms may have worsened after the COVID-19 lockdown was lifted [62,63]. In addition, concerns have been raised that when owners make decisions about COVID-19 testing and treatment, the presence and welfare of their pets may be one of the factors that delay their decisions [64]. Thus, under the COVID-19 lockdown, the health status of the pet and owner form a mutually influential relationship. In other words, there is a concern that a decline in the health of either the owner or the pet will negatively affect the health status of the other, creating a negative feedback loop. To be prepared for emergencies, pet owners need to maintain the health of both themselves and their pets at a high level on a daily basis.

This study also showed a correlation between owners’ low mental health and high levels of cat aggression towards owners and dogs. Cat aggression is one of the reasons for abandoning cats, as it decreases owner satisfaction and causes significant stress for the owner [57]. In other words, although this questionnaire showed no difference between before and after COVID-19, one hypothesis is that the increase in cat aggression may have caused a deterioration in the mental health of the owners. In fact, in Japan, the number of adult cats abandoned began to rise in 2020, even though it had been falling every year until 2019 [65]. While the reason is unclear due to a lack of detailed research, the COVID-19 lockdown may have been involved. Another hypothesis is that the owners’ deteriorating mental health may have led to increased cat aggression. For example, a recent study showed that owners with higher neurotic personality scores, which are known to be associated with various health problems, were more likely to adopt styles of interaction that cats did not like [66]. Owners, especially those with stronger attachments to cats, may perceive cats as a source of social support [67] and may rely on cats as a psychological stronghold. This is despite previous studies finding that under the COVID-19 lockdown, the presence of a pet does not act as a protective factor for owners with severe symptoms of mental health conditions and high attachment to their pets [68]. In other words, during times of emergency, the stress-related behaviours of cats and the health of their owners impact each other.

This study showed that dogs had more nonsocial fear and excitability and less trainability when their owners’ mental health was low. Although previous studies have shown that long-term stress states of dogs and owners are positively correlated [50,51], these results suggest that owners’ mental health is also correlated with stress-related behaviours in dogs. In addition, dogs are animals that exhibit obedient behavioural responses to their owners’ instructional commands. This trainability serves an important function in many animal-assisted therapies. For example, when dog training is incorporated into therapy programmes for prisoners or patients with PTSD, successful dog training enhances the subject’s sense of accomplishment and satisfaction [69,70]. In the general household, dog training is another form of communication that enhances owners’ sense of well-being [58,59]. However, men with moderate depression are more than five times more likely to use punitive training methods for dogs, which is also known to increase the prevalence and severity of problematic behaviours in dogs [71]. In other words, the deterioration of owners’ mental health and the decrease in dogs’ loyalty may interact and form a negative feedback loop. However, the aforementioned cat and dog behaviours did not change statistically significantly between before and after the COVID-19 lockdown. In fact, in contrast to cats, the number of abandoned adult dogs also decreased from 2019 to 2020 [65]. Therefore, the correlation between the effects of the COVID-19 pandemic, the mental health status of owners, and changes in pet behaviour still has room for discussion and warrants further study.

### 4.4. Limitations

Several limitations exist in this study. First, the survey respondents in this study had biases in demographic attributes. For example, the majority of survey respondents were high-salary earners and resided in the capital city of Tokyo (see Appendix A). Therefore, they were less likely to be psychologically and physiologically affected by COVID-19 because they were less likely to have been economically impacted by it. Further, major cities, including Tokyo, had longer COVID-19 lockdown periods than other regions [18]. In particular, it had the highest number of infected people [72] and is a region with stricter COVID-19 regulations [73]. Therefore, the results of this study could be considered to have extracted a representative sample for quantifying the impact of COVID-19 on people’s lives. In addition, the proportion of female respondents in this study was high. This gender bias is present in studies similar to this study [15]. However, Grigg et al.’s [74] survey on cat behaviour reported that the high proportion of female respondents was not a factor influencing the results of the study. Therefore, it is possible that the results of this study were also not influenced by gender bias. In any case, future studies should acquire and analyse larger sample sizes to reduce demographic attribute bias.

Second, this study was a questionnaire survey, and therefore only subjective evaluations by the owners were included. The advantages of questionnaires are that the human and time effort involved in the survey is small, thus making it easy to describe the psychological state of the respondents. However, a person’s health condition depends not only on the psychological aspect but also on the physiological aspect. Furthermore, it is difficult to quantify the physiological state of a person using only subjective methods. In addition, in the evaluation of pet behaviour, questionnaire surveys may be affected by owners’ evaluation bias. Indeed, owners have a reinforcement bias to evaluate their pets’ personality traits more favourably [75]. To eliminate this bias, objective evaluation methods such as the use of video data where pets’ behaviour can be analysed are recommended. However, these objective evaluation methods are often difficult to request from pet owners of ordinary households, which raises the concern that the sample size of the study will be small. Future studies should take a combined approach that incorporates the advantages of both subjective and objective evaluations.

Third, we conducted a cross-sectional, rather than longitudinal, study. In this study, we surveyed owners after the COVID-19 lockdown. Thus, owners assessed their own mental health at the time of response and retrospectively assessed their pet’s behaviour at the time of response and the pre-lockdown time period. Furthermore, the present study was conducted approximately six months after the COVID-19 lockdown occurred. This means that it is difficult to consider causal relationships in the results obtained in this study. Moreover, because of this method, we failed to describe changes before and after COVID-19, and most of our results did not show the effects before and after the spread of COVID-19 infection. Despite our best efforts, it took a long time to start the study because the questionnaire had to be prepared after the spread of infection and reviewed by the ethics committee (required time: 2.5 months). Ideally, we would have liked to have collected data before and after the pandemic. Japan is a country with a high incidence of disasters, including earthquakes and volcanic activity [76,77], which can easily cause changes in the lives of pet owners and their pets. The number of pets kept in Japan exceeds 15 million [24], and the impact of pets on Japanese society is immeasurable. It is worthwhile to study how emergencies such as natural disasters affect the lives and health of pets and their owners. For example, it would be beneficial to continuously monitor their health status, behaviour, and communication on a daily basis through surveys on a large scale by national and local governments. Future research should aim to establish such a system and thus conduct longitudinal studies.

The analyses conducted in this study were exploratory in nature and were not based on pre-registration with a hypothesis. Therefore, the results of the exhaustive analysis may be robust, but other results may include those that became significant by chance. Although it is unclear whether it will be possible in the future to acquire data in a similar emergency (which we hope to avoid if possible), to draw definitive conclusions, it will be necessary to formulate hypotheses based on the results of the present study and specify the analysis in advance before conducting the study.

## 5. Conclusions

This study investigated whether the COVID-19 lockdown affected pet behaviour, lifestyles, mental health, and owner-pet interaction in Japan. We found that an increase in owners’ time spent at home increased the frequency of contact interaction between owners and their pets and the attention-seeking behaviours of cats. However, increased contact interaction was also associated with increased stress-related behaviours in cats (excessive/compulsive self-grooming and fear) and dogs (aggression, fear, and touch sensitivity). Therefore, it is important for owners to communicate with their pets to suit their welfare and needs under the COVID-19 lockdown. In addition, owners with lower mental health scores tended to report more stress-related behaviours in cats (separation anxiety, touch sensitivity, and aggression) and dogs (separation anxiety, touch sensitivity, and fear). Owners and pets may strive to improve their mutual health and welfare on a daily basis to form a relationship where they can positively affect each other’s health status in times of emergency.

## Figures and Tables

**Figure 1 animals-13-02217-f001:**
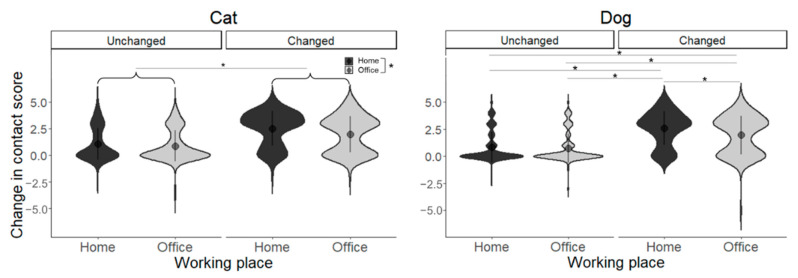
Change in the contact score for each group. Asterisks represent statistical significance *p* < 0.05.

**Figure 2 animals-13-02217-f002:**
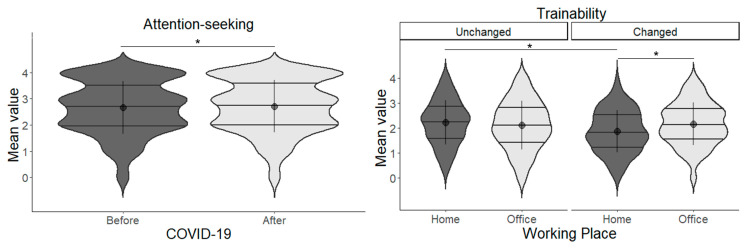
Relationship between the four life changes and the cats’ behaviour scored by Fe-BARQ. Asterisks represent statistical significance *p* < 0.05.

**Figure 3 animals-13-02217-f003:**
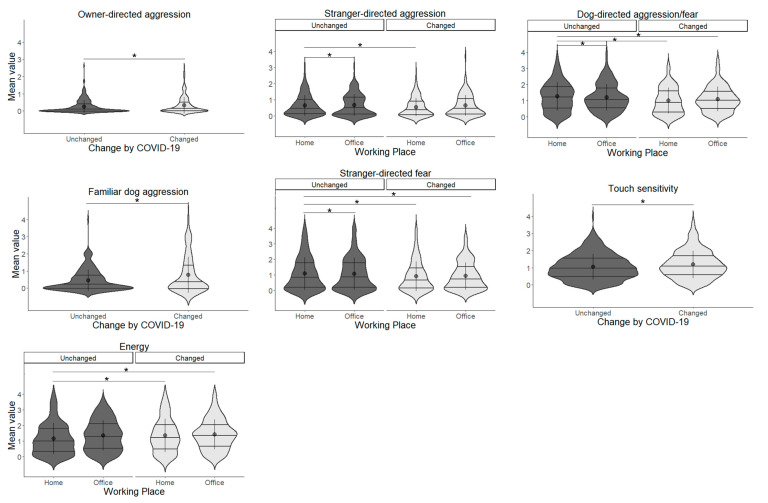
Relationship between the four life changes and the dogs’ behaviour scored by C-BARQ. Asterisks represent statistical significance *p* < 0.05.

**Figure 4 animals-13-02217-f004:**
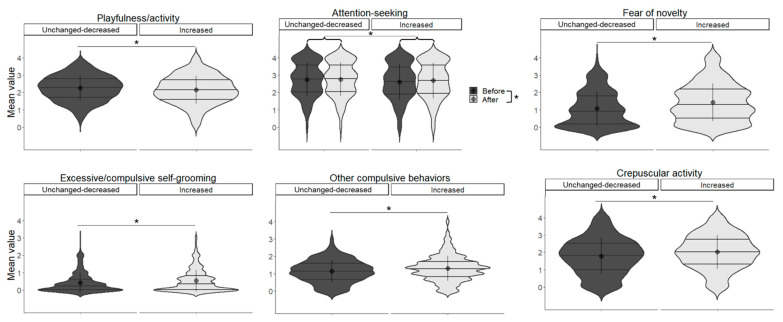
Relationship between the change in contact score and the cats’ behaviour scored by Fe-BARQ. Asterisks represent statistical significance *p* < 0.05.

**Figure 5 animals-13-02217-f005:**
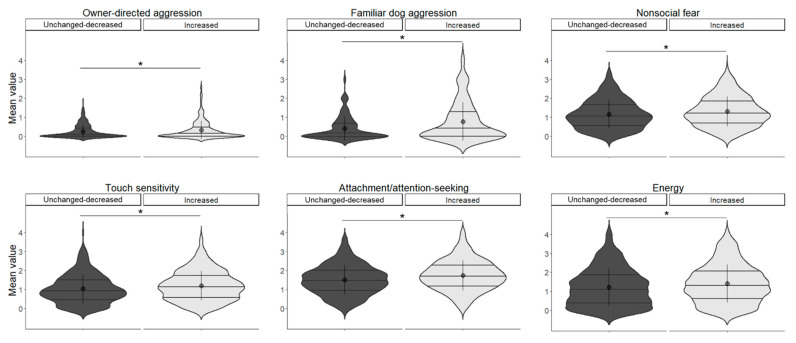
Relationship between the change in contact score and the dogs’ behaviour scored by C-BARQ. Asterisks represent statistical significance *p* < 0.05.

**Figure 6 animals-13-02217-f006:**
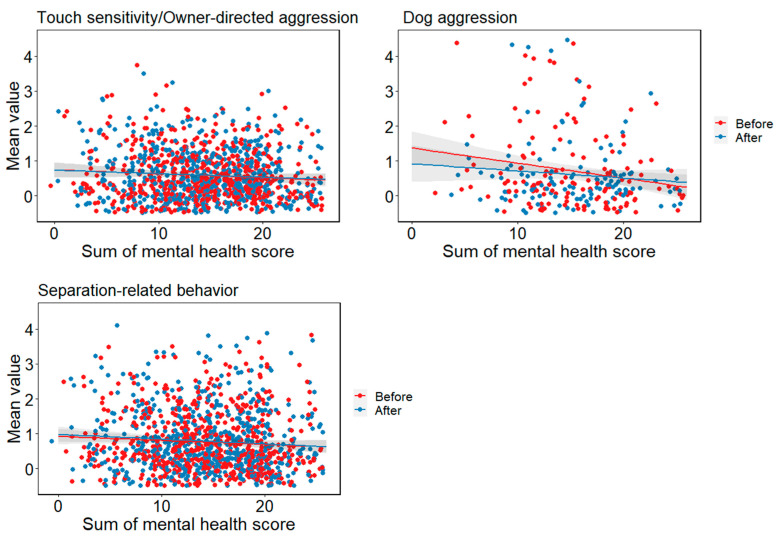
Relationship between the mental health of the owner and the cats’ behaviour scored by Fe-BARQ.

**Figure 7 animals-13-02217-f007:**
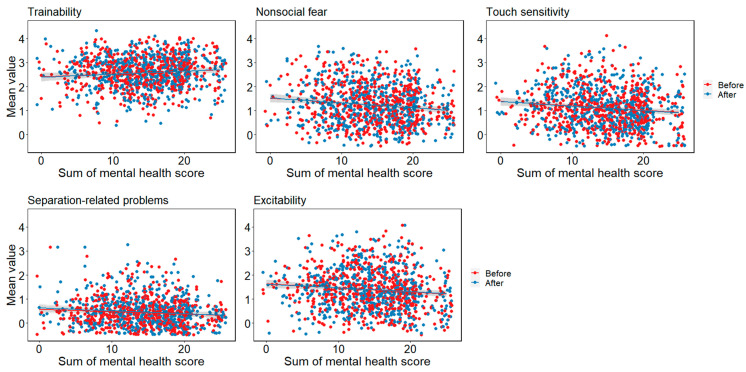
Relationship between the mental health of the owner and the dogs’ behaviour scored by C-BARQ.

**Table 1 animals-13-02217-t001:** Attribution of participants.

		Cat		Dog	
Items		No.	%	No.	%
**Gender**	Female	567	92.65	537	93.07
	Male	38	6.21	34	5.89
	Other/NA	7	1.14	6	1.04
	Total	612	100.00	577	100.00
**Number of cohabiting people**	0	145	23.69	90	15.6
	1	275	44.93	262	45.41
	2	121	19.77	145	25.13
	3	53	8.66	65	11.27
	4	17	2.78	13	2.25
	5	1	0.16	2	0.35
	Total	612	100.00	577	100.00
**Number of cohabiting cats or dogs**	0	279	45.59	433	75.04
	1	180	29.41	108	18.72
	2	85	13.89	27	4.68
	3	32	5.23	7	1.21
	4	14	2.29	0	0
	5	4	0.65	1	0.17
	6	7	1.14	1	0.17
	7	4	0.65	0	0
	8	3	0.49	0	0
	more than 9	4	0.65	0	0
	Total	612	100.00	577	100.00
**Changes in their working and schooling conditions due to the COVID-19 pandemic**	home/unchanged	131	21.41	161	27.9
	office/unchanged	183	29.9	169	29.29
	home/changed	142	23.2	109	18.89
	office/changed	156	25.49	138	23.92
	Total	612	100.00	577	100.00

**Table 2 animals-13-02217-t002:** Changes in each contact behaviour in the four life changes in COVID-19.

		Cat					Dog				
Behaviour Items	Changes	Home/ Unchanged	Office/ Unchanged	Home/ Changed	Office/ Changed	Total	Home/ Unchanged	Office/ Unchanged	Home/ Changed	Office/ Changed	Total
**Approaching**	**−1 (Decrease)**	1	2	2	2	7	0	1	0	1	2
	**0 (No Change)**	95	142	39	68	344	126	138	32	60	356
	**1 (Increase)**	35	39	101	86	261	35	30	77	77	219
**Petting**	**−1 (Decrease)**	1	2	2	1	6	0	1	0	1	2
	**0 (No change)**	93	128	39	68	328	123	135	28	62	348
	**1 (Increase)**	37	53	101	87	278	38	33	81	75	227
**Calling**	**−1 (Decrease)**	1	2	0	0	3	0	1	0	1	2
	**0 (No Change)**	83	124	40	66	313	125	131	28	56	340
	**1 (Increase)**	47	57	102	90	296	36	37	81	81	235
**Feeding**	**−1 (Decrease)**	2	1	1	2	6	1	2	1	1	5
	**0 (No Change)**	119	173	109	135	536	155	165	95	127	542
	**1 (Increase)**	10	9	32	19	70	5	2	13	10	30
**Giving treats**	**−1 (Decrease)**	2	6	2	0	10	4	2	4	3	13
	**0 (No Change)**	110	160	107	122	499	133	139	71	101	444
	**1 (Increase)**	19	17	33	34	103	24	28	34	34	120
**Change in contact score**	** *M* **	1.08	0.89	2.55	1.99	1.59	0.83	0.73	2.58	1.96	1.40
	** *SD* **	1.51	1.46	1.6	1.73	1.71	1.4	1.35	1.55	1.77	1.68
**Change in contact group**	**Unchanged-** **Decreased**	75	113	26	53	267	106	115	20	45	286
	**Increased**	56	70	116	103	345	55	54	89	93	291

**Table 3 animals-13-02217-t003:** Dogs’ walking frequency and duration.

		Home/ Unchanged		Office/ Unchanged		Home/ Changed		Office/ Changed		Total	
Items		before	after	before	after	before	after	before	after	before	after
**Walking Frequency**	**Twice a day**	81	82	84	81	50	55	64	66	279	284
	**Once a day**	51	43	43	45	30	27	48	46	172	161
	**Two to four times a week**	12	16	21	21	13	16	15	15	61	68
	**Once a week**	4	6	8	6	8	4	3	1	23	17
	**Two to three times a month**	2	1	2	2	3	3	3	3	10	9
	**Hardly ever**	7	8	5	8	3	2	4	5	19	23
	**Other**	4	5	6	6	2	2	1	2	13	15
**Walking Duration**	**More than an hour**	31	29	36	36	13	18	28	26	108	109
	**30 min to an hour**	78	73	78	72	46	48	65	70	267	263
	**15 to 30 min**	33	40	38	40	34	29	33	30	138	139
	**Less than 15 min**	19	19	17	21	16	14	12	12	64	66

**Table 4 animals-13-02217-t004:** Mental health in four life changes in COVID-19.

		Home/ Unchanged	Office/ Unchanged	Home/ Changed	Office/ Changed	Total
**Cat**	** *M* **	14.14	14.31	14.76	14.75	14.49
	** *SD* **	5.60	5.78	4.60	5.07	5.30
**Dog**	** *M* **	13.97	14.50	15.13	14.57	14.49
	** *SD* **	5.48	5.91	4.62	5.25	5.41

**Table 5 animals-13-02217-t005:** Physical activity in four life changes in COVID-19.

		Home/ Unchanged	Office/ Unchanged	Home/ Changed	Office/ Changed	Total
**Cat**	** *M* **	8.11	8.15	6.90	8.57	7.96
	** *SD* **	6.79	7.19	5.60	6.96	6.72
**Dog**	** *M* **	10.46	10.98	11.07	10.08	10.64
	** *SD* **	7.27	8.18	7.87	6.98	7.59

**Table 6 animals-13-02217-t006:** Changes in dog walking frequency and duration.

Items			Home/ Unchanged	Office/ Unchanged	Home/ Changed	Office/ Changed	Total
**Frequency**	**before**	** *M* **	4.17	4.13	4.00	4.13	4.12
		** *SD* **	1.23	1.20	1.27	1.15	1.21
	**after**	** *M* **	4.12	4.06	4.13	4.15	4.11
		** *SD* **	1.29	1.29	1.17	1.18	1.24
**Duration**	**before**	** *M* **	2.75	2.79	2.51	2.79	2.73
		** *SD* **	0.90	0.89	0.89	0.87	0.89
	**after**	** *M* **	2.70	2.73	2.64	2.80	2.72
		** *SD* **	0.90	0.94	0.91	0.85	0.90

**Table 7 animals-13-02217-t007:** Fe-BARQ score in four life changes in COVID-19.

Items			Home/ Unchanged	Office/ Unchanged	Home/ Changed	Office/ Changed	Total
**Playfulness/activity**	**before**	** *M* **	2.18	2.21	2.16	2.24	2.20
		** *SD* **	0.77	0.77	0.78	0.74	0.77
	**after**	** *M* **	2.11	2.13	2.12	2.21	2.14
		** *SD* **	0.81	0.81	0.82	0.77	0.80
**Sociability**	**before**	** *M* **	0.95	1.21	1.15	1.17	1.13
		** *SD* **	1.15	1.25	1.18	1.21	1.20
	**after**	** *M* **	0.89	1.15	1.05	1.09	1.06
		** *SD* **	1.16	1.24	1.25	1.18	1.21
**Directed calls/vocalizations**	**before**	** *M* **	2.83	2.69	2.58	2.71	2.70
		** *SD* **	0.85	0.82	0.83	0.89	0.85
	**after**	** *M* **	2.78	2.60	2.62	2.67	2.66
		** *SD* **	0.95	0.94	0.98	0.96	0.96
**Purring**	**before**	** *M* **	3.20	3.10	3.01	3.15	3.11
		** *SD* **	0.79	0.90	0.98	0.96	0.91
	**after**	** *M* **	3.20	3.10	3.05	3.15	3.12
		** *SD* **	0.81	0.91	0.94	0.97	0.91
**Attention-seeking**	**before**	** *M* **	2.77	2.68	2.52	2.70	2.67
		** *SD* **	1.03	1.02	0.97	1.00	1.01
	**after**	** *M* **	2.81	2.73	2.62	2.71	2.72
		** *SD* **	1.03	1.00	0.99	0.99	1.00
**Sociability with cats**	**before**	** *M* **	0.58	0.80	0.95	0.63	0.74
		** *SD* **	0.96	0.98	1.32	1.02	1.07
	**after**	** *M* **	0.41	0.49	0.51	0.35	0.44
		** *SD* **	0.84	0.83	1.04	0.79	0.86
**Stranger-directed aggression**	**before**	** *M* **	0.94	0.78	0.90	0.86	0.86
		** *SD* **	1.26	1.14	1.21	1.20	1.19
	**after**	** *M* **	0.75	0.65	0.67	0.64	0.67
		** *SD* **	1.27	1.12	1.13	1.18	1.17
**Touch sensitivity/** **Owner-directed aggression**	**before**	** *M* **	0.50	0.58	0.67	0.56	0.58
		** *SD* **	0.69	0.61	0.65	0.65	0.65
	**after**	** *M* **	0.38	0.46	0.55	0.44	0.46
		** *SD* **	0.60	0.57	0.63	0.59	0.60
**Resistance to restraint**	**before**	** *M* **	0.72	0.81	0.81	0.85	0.80
		** *SD* **	0.89	1.00	0.90	0.94	0.94
	**after**	** *M* **	0.68	0.81	0.78	0.84	0.78
		** *SD* **	0.88	1.04	0.88	0.98	0.96
**Familiar cat aggression**	**before**	** *M* **	0.53	0.35	0.44	0.36	0.41
		** *SD* **	0.76	0.57	0.69	0.70	0.67
	**after**	** *M* **	0.55	0.42	0.60	0.47	0.50
		** *SD* **	0.80	0.70	0.82	0.81	0.77
**Dog aggression**	**before**	** *M* **	0.89	0.66	0.82	0.53	0.72
		** *SD* **	1.22	0.89	1.29	0.89	1.06
	**after**	** *M* **	0.67	0.66	0.67	0.38	0.59
		** *SD* **	1.15	0.93	1.11	0.64	0.96
**Fear of unfamiliar dogs/cats**	**before**	** *M* **	0.98	0.88	0.98	1.33	1.03
		** *SD* **	1.36	1.29	1.42	1.44	1.37
	**after**	** *M* **	0.94	0.83	0.96	1.16	0.95
		** *SD* **	1.36	1.27	1.39	1.42	1.35
**Fear of novelty**	**before**	** *M* **	1.27	1.17	1.39	1.29	1.27
		** *SD* **	1.04	1.05	1.05	1.10	1.06
	**after**	** *M* **	1.29	1.16	1.39	1.30	1.28
		** *SD* **	1.04	1.05	1.08	1.11	1.07
**Separation-related behaviour**	**before**	** *M* **	0.79	0.73	0.81	0.71	0.76
		** *SD* **	0.82	0.81	0.81	0.77	0.80
	**after**	** *M* **	0.69	0.68	0.77	0.64	0.69
		** *SD* **	0.79	0.80	0.87	0.79	0.81
**Trainability**	**before**	** *M* **	2.22	2.10	1.83	2.15	2.08
		** *SD* **	0.90	0.98	0.87	0.86	0.92
	**after**	** *M* **	2.00	1.91	1.66	1.96	1.88
		** *SD* **	1.02	1.10	1.02	0.95	1.03
**Predatory behaviour**	**before**	** *M* **	1.73	1.26	1.52	1.39	1.46
		** *SD* **	1.35	1.15	1.21	1.18	1.23
	**after**	** *M* **	2.11	1.81	1.95	1.75	1.90
		** *SD* **	1.52	1.37	1.28	1.42	1.40
**Prey interest**	**before**	** *M* **	2.08	2.01	1.81	2.08	1.99
		** *SD* **	1.04	1.07	1.04	1.14	1.08
	**after**	** *M* **	2.00	1.97	1.79	2.08	1.96
		** *SD* **	1.10	1.09	1.06	1.15	1.10
**Location preferences for resting/** **sleeping**	**before**	** *M* **	1.65	1.84	1.76	1.71	1.75
		** *SD* **	0.94	0.93	0.96	0.94	0.94
	**after**	** *M* **	1.25	1.51	1.41	1.38	1.40
		** *SD* **	1.01	1.05	1.11	1.04	1.05
**Excessive/** **compulsive self-grooming**	**before**	** *M* **	0.52	0.39	0.48	0.51	0.47
		** *SD* **	0.65	0.53	0.59	0.61	0.59
	**after**	** *M* **	0.43	0.30	0.43	0.41	0.39
		** *SD* **	0.60	0.48	0.59	0.62	0.57
**Other compulsive behaviours**	**before**	** *M* **	1.21	1.15	1.30	1.24	1.22
		** *SD* **	0.67	0.65	0.72	0.69	0.68
	**after**	** *M* **	0.92	0.83	0.97	0.89	0.90
		** *SD* **	0.79	0.72	0.84	0.76	0.77
**Inappropriate elimination**	**before**	** *M* **	0.30	0.32	0.35	0.31	0.32
		** *SD* **	0.56	0.64	0.66	0.66	0.63
	**after**	** *M* **	0.26	0.31	0.33	0.28	0.30
		** *SD* **	0.55	0.63	0.66	0.68	0.63
**Elimination preferences**	**before**	** *M* **	1.69	1.41	1.39	1.36	1.45
		** *SD* **	1.50	1.42	1.47	1.46	1.46
	**after**	** *M* **	1.67	1.40	1.37	1.36	1.44
		** *SD* **	1.50	1.42	1.48	1.45	1.46
**Crepuscular activity**	**before**	** *M* **	1.85	1.84	2.02	1.91	1.90
		** *SD* **	0.99	1.01	1.03	1.02	1.01
	**after**	** *M* **	1.90	1.89	2.02	1.91	1.93
		** *SD* **	1.13	1.18	1.15	1.20	1.17
**Other behaviour**	**before**	** *M* **	1.22	1.27	1.22	1.22	1.24
		** *SD* **	0.40	0.47	0.42	0.41	0.43
	**after**	** *M* **	1.22	1.27	1.23	1.24	1.24
		** *SD* **	0.43	0.48	0.43	0.41	0.44

**Table 8 animals-13-02217-t008:** C-BARQ score in four life changes in COVID-19.

Items			Home/ Unchanged	Office/ Unchanged	Home/ Changed	Office/ Changed	Total
**Trainability**	**before**	** *M* **	2.60	2.61	2.58	2.52	2.58
		** *SD* **	0.58	0.57	0.58	0.62	0.59
	**after**	** *M* **	2.62	2.61	2.59	2.56	2.60
		** *SD* **	0.57	0.59	0.62	0.61	0.60
**Owner-directed aggression**	**before**	** *M* **	0.20	0.29	0.36	0.32	0.28
		** *SD* **	0.34	0.38	0.54	0.44	0.42
	**after**	** *M* **	0.20	0.29	0.33	0.31	0.28
		** *SD* **	0.35	0.40	0.49	0.47	0.42
**Stranger-directed aggression**	**before**	** *M* **	0.64	0.65	0.55	0.64	0.63
		** *SD* **	0.65	0.65	0.59	0.65	0.64
	**after**	** *M* **	0.67	0.70	0.56	0.65	0.65
		** *SD* **	0.67	0.69	0.59	0.67	0.66
**Dog-directed aggression/fear**	**before**	** *M* **	1.25	1.20	0.99	1.11	1.15
		** *SD* **	0.89	0.84	0.80	0.76	0.83
	**after**	** *M* **	1.30	1.25	1.03	1.12	1.19
		** *SD* **	0.89	0.85	0.83	0.79	0.85
**Chasing**	**before**	** *M* **	1.39	1.42	1.40	1.54	1.44
		** *SD* **	1.10	1.07	1.23	1.08	1.11
	**after**	** *M* **	1.46	1.51	1.52	1.64	1.53
		** *SD* **	1.21	1.24	1.43	1.30	1.28
**Familiar dog aggression**	**before**	** *M* **	0.39	0.48	0.72	0.80	0.58
		** *SD* **	0.55	0.61	1.15	1.01	0.84
	**after**	** *M* **	0.47	0.39	0.79	0.85	0.61
		** *SD* **	0.68	0.56	1.17	1.10	0.91
**Stranger-directed fear**	**before**	** *M* **	1.07	1.06	0.93	0.93	1.01
		** *SD* **	1.06	1.03	0.98	0.89	1.00
	**after**	** *M* **	1.10	1.07	0.90	0.95	1.02
		** *SD* **	1.10	1.03	0.93	0.88	1.00
**Nonsocial fear**	**before**	** *M* **	1.26	1.19	1.30	1.22	1.23
		** *SD* **	0.83	0.76	0.76	0.72	0.77
	**after**	** *M* **	1.13	1.02	1.16	1.02	1.07
		** *SD* **	0.81	0.76	0.78	0.70	0.76
**Touch sensitivity**	**before**	** *M* **	1.04	1.07	1.32	1.13	1.13
		** *SD* **	0.75	0.74	0.85	0.74	0.77
	**after**	** *M* **	0.81	0.79	0.99	0.85	0.85
		** *SD* **	0.76	0.74	0.89	0.77	0.78
**Separation-related problems**	**before**	** *M* **	0.45	0.40	0.46	0.42	0.43
		** *SD* **	0.60	0.45	0.53	0.45	0.51
	**after**	** *M* **	0.49	0.46	0.51	0.47	0.48
		** *SD* **	0.63	0.53	0.61	0.51	0.57
**Excitability**	**before**	** *M* **	1.32	1.41	1.43	1.41	1.39
		** *SD* **	0.70	0.86	0.83	0.79	0.79
	**after**	** *M* **	1.38	1.44	1.46	1.41	1.42
		** *SD* **	0.73	0.88	0.84	0.82	0.82
**Attachment/attention-seeking**	**before**	** *M* **	1.51	1.60	1.72	1.60	1.60
		** *SD* **	0.81	0.78	0.78	0.70	0.77
	**after**	** *M* **	1.58	1.67	1.83	1.66	1.67
		** *SD* **	0.84	0.83	0.83	0.76	0.82
**Energy**	**before**	** *M* **	1.16	1.36	1.35	1.44	1.32
		** *SD* **	0.99	0.97	1.07	0.96	1.00
	**after**	** *M* **	1.15	1.36	1.36	1.39	1.31
		** *SD* **	1.00	0.98	1.07	0.95	1.00

**Table 9 animals-13-02217-t009:** Fe-BARQ score change in the contact group.

Items			Unchanged- Decreased	Increased	Total
**Playfulness/activity**	**before**	** *M* **	2.25	2.16	2.20
		** *SD* **	0.73	0.80	0.77
	**after**	** *M* **	2.19	2.11	2.14
		** *SD* **	0.75	0.83	0.80
**Sociability**	**before**	** *M* **	1.08	1.17	1.13
		** *SD* **	1.17	1.22	1.20
	**after**	** *M* **	1.01	1.10	1.06
		** *SD* **	1.16	1.25	1.21
**Directed calls/vocalizations**	**before**	** *M* **	2.74	2.67	2.70
		** *SD* **	0.80	0.88	0.85
	**after**	** *M* **	2.70	2.63	2.66
		** *SD* **	0.94	0.97	0.96
**Purring**	**before**	** *M* **	3.15	3.09	3.11
		** *SD* **	0.91	0.91	0.91
	**after**	** *M* **	3.14	3.11	3.12
		** *SD* **	0.92	0.91	0.91
**Attention-seeking**	**before**	** *M* **	2.74	2.61	2.67
		** *SD* **	0.97	1.03	1.01
	**after**	** *M* **	2.76	2.68	2.72
		** *SD* **	0.96	1.03	1.00
**Sociability with cats**	**before**	** *M* **	0.73	0.74	0.74
		** *SD* **	1.03	1.10	1.07
	**after**	** *M* **	0.44	0.45	0.44
		** *SD* **	0.85	0.87	0.86
**Stranger-directed aggression**	**before**	** *M* **	0.90	0.83	0.86
		** *SD* **	1.25	1.15	1.19
	**after**	** *M* **	0.70	0.65	0.67
		** *SD* **	1.24	1.10	1.17
**Touch sensitivity/owner-directed aggression**	**before**	** *M* **	0.56	0.59	0.58
		** *SD* **	0.66	0.64	0.65
	**after**	** *M* **	0.41	0.49	0.46
		** *SD* **	0.59	0.60	0.60
**Resistance to restraint**	**before**	** *M* **	0.78	0.82	0.80
		** *SD* **	0.98	0.91	0.94
	**after**	** *M* **	0.76	0.79	0.78
		** *SD* **	1.02	0.90	0.96
**Familiar cat aggression**	**before**	** *M* **	0.35	0.47	0.41
		** *SD* **	0.62	0.72	0.67
	**after**	** *M* **	0.39	0.60	0.50
		** *SD* **	0.70	0.83	0.77
**Dog aggression**	**before**	** *M* **	0.82	0.64	0.72
		** *SD* **	1.02	1.09	1.06
	**after**	** *M* **	0.72	0.48	0.59
		** *SD* **	0.91	1.00	0.96
**Fear of unfamiliar dogs/cats**	**before**	** *M* **	1.17	0.91	1.03
		** *SD* **	1.46	1.29	1.37
	**after**	** *M* **	1.12	0.80	0.95
		** *SD* **	1.42	1.26	1.35
**Fear of novelty**	**before**	** *M* **	1.07	1.43	1.27
		** *SD* **	1.00	1.08	1.06
	**after**	** *M* **	1.07	1.43	1.28
		** *SD* **	1.01	1.10	1.07
**Separation-related behaviour**	**before**	** *M* **	0.72	0.78	0.76
		** *SD* **	0.82	0.79	0.80
	**after**	** *M* **	0.65	0.72	0.69
		** *SD* **	0.80	0.82	0.81
**Trainability**	**before**	** *M* **	2.08	2.07	2.08
		** *SD* **	0.90	0.93	0.92
	**after**	** *M* **	1.85	1.91	1.88
		** *SD* **	1.03	1.04	1.03
**Predatory behaviour**	**before**	** *M* **	1.50	1.42	1.46
		** *SD* **	1.29	1.18	1.23
	**after**	** *M* **	1.98	1.84	1.90
		** *SD* **	1.39	1.40	1.40
**Prey interest**	**before**	** *M* **	2.03	1.96	1.99
		** *SD* **	1.07	1.09	1.08
	**after**	** *M* **	2.02	1.92	1.96
		** *SD* **	1.10	1.10	1.10
**Location preferences for resting/sleeping**	**before**	** *M* **	1.80	1.71	1.75
		** *SD* **	0.98	0.91	0.94
	**after**	** *M* **	1.44	1.37	1.40
		** *SD* **	1.05	1.06	1.05
**Excessive/compulsive self-grooming**	**before**	** *M* **	0.42	0.51	0.47
		** *SD* **	0.56	0.61	0.59
	**after**	** *M* **	0.32	0.44	0.39
		** *SD* **	0.50	0.62	0.57
**Other compulsive behaviours**	**before**	** *M* **	1.13	1.29	1.22
		** *SD* **	0.64	0.71	0.68
	**after**	** *M* **	0.82	0.95	0.90
		** *SD* **	0.70	0.82	0.77
**Inappropriate elimination**	**before**	** *M* **	0.28	0.35	0.32
		** *SD* **	0.56	0.69	0.63
	**after**	** *M* **	0.26	0.33	0.30
		** *SD* **	0.56	0.69	0.63
**Elimination preferences**	**before**	** *M* **	1.35	1.53	1.45
		** *SD* **	1.44	1.48	1.46
	**after**	** *M* **	1.35	1.51	1.44
		** *SD* **	1.44	1.48	1.46
**Crepuscular activity**	**before**	** *M* **	1.78	2.00	1.90
		** *SD* **	1.05	0.98	1.01
	**after**	** *M* **	1.78	2.04	1.93
		** *SD* **	1.16	1.16	1.17
**Other behaviour**	**before**	** *M* **	1.23	1.24	1.24
		** *SD* **	0.44	0.42	0.43
	**after**	** *M* **	1.24	1.25	1.24
		** *SD* **	0.45	0.43	0.44

**Table 10 animals-13-02217-t010:** C-BARQ score change in the contact group.

Items			Unchanged- Decreased	Increased	Total
**Trainability**	**before**	** *M* **	2.62	2.54	2.58
		** *SD* **	0.59	0.58	0.59
	**after**	** *M* **	2.62	2.57	2.60
		** *SD* **	0.60	0.59	0.60
**Owner-directed aggression**	**before**	** *M* **	0.25	0.32	0.28
		** *SD* **	0.35	0.48	0.42
	**after**	** *M* **	0.24	0.32	0.28
		** *SD* **	0.36	0.48	0.42
**Stranger-directed aggression**	**before**	** *M* **	0.59	0.67	0.63
		** *SD* **	0.63	0.64	0.64
	**after**	** *M* **	0.61	0.70	0.65
		** *SD* **	0.65	0.66	0.66
**Dog-directed aggression**	**before**	** *M* **	1.13	1.17	1.15
		** *SD* **	0.87	0.79	0.83
	**after**	** *M* **	1.18	1.20	1.19
		** *SD* **	0.89	0.80	0.85
**Chasing**	**before**	** *M* **	1.40	1.47	1.44
		** *SD* **	1.07	1.15	1.11
	**after**	** *M* **	1.48	1.57	1.53
		** *SD* **	1.22	1.33	1.28
**Familiar dog aggression**	**before**	** *M* **	0.43	0.74	0.58
		** *SD* **	0.64	0.98	0.84
	**after**	** *M* **	0.42	0.82	0.61
		** *SD* **	0.70	1.05	0.91
**Stranger-directed fear**	**before**	** *M* **	0.98	1.04	1.01
		** *SD* **	0.96	1.03	1.00
	**after**	** *M* **	0.97	1.06	1.02
		** *SD* **	0.97	1.02	1.00
**Nonsocial fear**	**before**	** *M* **	1.16	1.30	1.23
		** *SD* **	0.75	0.78	0.77
	**after**	** *M* **	0.99	1.16	1.07
		** *SD* **	0.73	0.79	0.76
**Touch sensitivity**	**before**	** *M* **	1.03	1.21	1.13
		** *SD* **	0.76	0.77	0.77
	**after**	** *M* **	0.75	0.94	0.85
		** *SD* **	0.76	0.80	0.78
**Separation-related problems**	**before**	** *M* **	0.40	0.46	0.43
		** *SD* **	0.53	0.49	0.51
	**after**	** *M* **	0.44	0.52	0.48
		** *SD* **	0.57	0.57	0.57
**Excitability**	**before**	** *M* **	1.33	1.44	1.39
		** *SD* **	0.80	0.78	0.79
	**after**	** *M* **	1.35	1.48	1.42
		** *SD* **	0.82	0.81	0.82
**Attachment/attention-seeking**	**before**	** *M* **	1.49	1.70	1.60
		** *SD* **	0.76	0.77	0.77
	**after**	** *M* **	1.54	1.80	1.67
		** *SD* **	0.80	0.82	0.82
**Energy**	**before**	** *M* **	1.22	1.42	1.32
		** *SD* **	0.99	1.00	1.00
	**after**	** *M* **	1.22	1.40	1.31
		** *SD* **	1.00	1.00	1.00

## Data Availability

The data presented in this study are not available because of the restrictions on the content of the ethics application.

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
