# Peer review of "Effects of the COVID-19 Pandemic on the Behavioural Tendencies of Cats and Dogs in Japan"

_animals, 2023, doi:10.3390/ani13132217_

Round 1

Reviewer 1 Report

Dear Authors,

Neither substantively nor methodologically, there is nothing to complain about. The work also deals with interesting topics. In my opinion, however, it needs a thorough shortening.

For example

line 75-87: In my opinion, this fragment is redundant, especially since the work is already such extensive

line 119-193: It is worth considering whether simply placing the questionnaire in the supplementary materials does not allow for shortening this part. I think a description of the essential parts of the questionnaire regarding the supplementary files for details would suffice

Author Response

Thank you very much for your time for reviewing our paper. Please see the attachment.

Reviewer 2 Report

General comments

The present study addresses an important and interesting topic regarding the COVID-19 pandemic and the interaction between companion animals and their owners. It provides an extensive study of animals' and humans' responses to the lockdown in Japan. I have left some comments, particularly about stating the aim of the study and clarifying the statistical analysis.

Line 25: Before stating that the authors conducted an online questionnaire, I recommend adding the aim of the study, included in lines 103-105.

Response:

Line 36: A short introduction about why it is important to evaluate the effect of COVID-19 on human-animal interaction is advised.

Response:

Lines 36-37: I recommend homologating the aim provided in these lines and the one stated in lines 103-105.

Response:

Line 40: Please, define the abbreviation for Fe-BARQ and C-BARQ.

Response:

Line 105: Consider adding a hypothesis for the present study.

Response:

Line 106: Please remove the extra spaces before Fe-BARQ.

Response:

Lines 113-118: The participants identified themselves with an official document where their age appeared, and you were able in some way to verify that they had pets at home and the veracity of what they said? I consider that it would be important to clarify this to know the real socioeconomic level, age, and level of education of the respondents to define if this study could be classified within a sector of the population or various. The bond and relationship that the owner has with his pet can vary greatly depending on his pace of life and socioeconomic level. Also, if there were other inclusion criteria or a method by which the participants were selected, please, explain them.

Response:

Line 116- 117: The sentence “Six hundred and twelve cat owners”. Regularly, numbers below ten are used to be written with letters, and above ten with numbers.

Response:

Line 119: Could you please attach a copy of the questionnaire that was applied?

Response:

Lines 135-137: What was the importance of collecting information about the coat color, eye color, and hair length of the cats?

Response:

Lines 163-164: Please, clearly define the 14 items to assess mobility, strength, and balance.

Response:

Line 169: Consider modifying the title of the subtopic to “Evaluation of the human-animal interaction frequency before and after the COVID-19 pandemic”.

Response:

Lines 177- 180: Here, I consider it adequate to cite the Fe-BARQ and C-BARQ or to briefly explain if these questionnaires assess the observation frequency per day or during the whole lockdown.

Response:

Line 187: Delete the extra space.

Response:

Line 230: Please, delete the extra spaces after “COVID”.

Response:

Line 234: Please, specify what type of ANOVA was used and the type of included effects (e.g., per animal or for the human-animal interaction).

Response:

Line 273 Results section: I consider that an important aspect to evaluate and that apparently, they did not consider is whether the food consumption and the weight of the animals evaluated increased during COVID. In my own experience, I have seen that the patients increased their food consumption and their weight when not having leisure time outdoors (which increased leisure time and decreased physical activity). In case you have considered that aspect I believe that this information could enrich your document.

Response:

Line 293: Please, add “average age” when mentioning 48.41 years.

Response:

Line 337: Please add the missing D for “COVI”.

Response:

Line 438: Add a space between “was” and “higher”.

Response:

Lines 547-548:  Which countries are the authors referring to?

Response:

Lines 551-552: Please, state two or three health benefits that companion animals can provide to owners.

Response:

Line 650-661: Here, the authors could mention other studies regarding dog aggression (e.g., 10.3390/children8080620) and the importance of preventing anthropomorphizing companion animals (e.g., https://doi.org/10.3390/ani11113263).

Response:

Lines 693-695: It would be interesting to mention the number of abandoned dogs and cats due to the COVID-19 pandemic.

Response:

Author Response

Thank you for your time and for providing us with many valuable comments. Please see the attachment.

Round 2

Reviewer 1 Report

for me it’s ok now

Reviewer 2 Report

The authors have responded to each of my comments.

I am satisfied with the improvements to the paper.

The manuscript can be published now.